# DISSECTING LANGUAGE MODELS:
# MACHINE UNLEARNING VIA SELECTIVE PRUNING

## ABSTRACT

Understanding and shaping the behaviour of Large Language Models (LLMs) is increasingly important as applications become more powerful and more frequently adopted. This paper introduces a machine unlearning method specifically designed for LLMs. We introduce a *selective pruning* method for LLMs that removes neurons based on their relative importance on a targeted capability compared to overall network performance. This approach is a compute- and data-efficient method for identifying and removing neurons that enable specific behaviours. Our findings reveal that both feed-forward and attention neurons in LLMs are specialized; that is, for specific tasks, certain neurons are more crucial than others.

## 1    INTRODUCTION

In the last two years, Large Language Models (LLMs) have been shown to achieve impressive performance in a wide array of skills. These skills have huge potential to create significant benefits for humanity. However, certain abilities may carry inherent risks, both through wide access to powerful models enabling misuse by bad actors, and from misalignment of the model's goals to its user's. Elimination of high-risk skills from an LLMs repertoire could be a valuable precaution against both misuse and misalignment. Additionally, datasets may contain sensitive user information or copyrighted material (where consent was not obtained or withdrawn) which should be removed.

Machine unlearning is a field that focuses on forgetting ability on one dataset while maintaining ability on a retain dataset. In recent years a wide variety of approaches has sprung up (Nguyen et al., 2022b), (Golatkar et al., 2020). However, there are challenges in applying these methods to LLMs, since a forward or backward pass in an LLM is costly (Bender et al., 2021).

In this paper, we introduce a method we call *selective pruning*. We evaluate our method by selectively removing coding ability in LLMs. Coding was chosen due to it being a common and powerful skill with excellent datasets, but with limited risk in research settings. Our proposed method is task-agnostic, requiring only a small dataset representative of the target task, and thus, we anticipate their applicability in the removal of other potentially harmful skills, such as manipulation. Selective pruning demands only a very small amount of additional data and computational resources.

A secondary aim of this research is to gain a deeper understanding of how various abilities are interconnected within LLMs. Our aim is separability rather than sparsity per se, which is why, contrary to most pruning methods (Blalock et al., 2020), we investigated pruning neurons (structured pruning) rather than pruning weights. If capabilities can be separated on the level of neurons, then this can lead to modularity inside models. We find that certain neurons are task-specialized, and removing them dramatically decreases performance on the forget dataset while hardly affecting performance on the retain dataset.

## 2    RELATED WORK

**Machine unlearning** aims to selectively remove information corresponding to specific data points without retraining the entire model from scratch. It has applications in for example privacy protection; complying with regulations such as GDPR; and in removing outdated or incorrect information from a trained model (Bourtoule et al., 2021). Typically, machine unlearning refers to removing the impact that a specific datapoint in the training set had on the final model (Tarun et al., 2021; Thudi et al.,

2022; Zhang et al., 2022c; Kurmanji et al., 2023). By this term we instead mean 'removing ability on a dataset that exemplifies certain behaviour (such as toxicity) or a certain skill (such as coding)'.

Certain machine unlearning methods geared towards neural networks are impractical to apply to LLMs. Jia et al. (2023) report fisher forgetting, influence unlearning, and fine-tuning and gradient ascent based machine unlearning methods cost between 2% and 9% the cost of retraining a model, for large language models this is quite expensive. For example DeltaGrad requires storing updates based on single data items during training (Nguyen et al., 2022b), which is costly for LLMs. We instead propose a post-hoc model surgery approach, in which we calculate influence functions after training. Ma et al. (2022) introduce a technique performing neuron masking in neural networks, but focus on unlearning specific data points, use gradient-based update techniques and do not focus on LLMs. Foster et al. (2023) unlearn classes from vision transformers using selective synaptic dampening.

**Behavioural control.** Reinforcement Learning from Human Feedback (RLHF) Christiano et al. (2017) can suppress behaviour, but does not eradicate knowledge, as observed through adversarial prompting and jailbreaking (Deng et al., 2023). Gandikota et al. (2023) erase concepts from text-to-image diffusion models by editing model weights. A very recent approach to LLM behaviour control is activation engineering. Turner et al. (2023) introduce a method that adds a vector to activations to control the model outputs.

**Pruning.** ACDC is a pruning-based approach to find a sub-circuit responsible for performance on a specific dataset (Conmy et al., 2023). This method aims to automate a step in the mechanistic interpretability pipeline. ACDC is related to selective pruning in that it uses pruning on LLMs, but the method prunes weights rather than neurons and has a very different application.

Neural network pruning typically focuses on retaining capabilities with a smaller compute budget. Networks are made sparser to e.g. reduce the storage footprint of the network, the computational cost of inference, or the energy requirements of inference (Blalock et al., 2020). For example, Michel et al. (2019) prune unused attention heads without significantly impacting performance. In contrast, we prune with the *aim* of *selectively* reducing performance.

## 3 SELECTIVE PRUNING

We introduce a machine unlearning method for Transformer models. Our method performs structured pruning to a trained LLM to selectively remove capabilities from the model. We either iteratively prune nodes in the feed-forward layers or attention head layers. We additionally show the method generalises to Vision Transformers.

The task or dataset that we aim to reduce performance on is referred to as the *forget* dataset ($D_{\text{forget}}$) and the task that we are optimizing for as the *retain* dataset ($D_{\text{retain}}$). Our method is a heuristic pruning technique and we selectively prune nodes based on their relative importance to the $D_{\text{forget}}$ and $D_{\text{retain}}$ datasets. A scoring function is used to determine which nodes to prune.

### 3.1 IMPORTANCE FUNCTIONS AND SCORING

We notice that zero is a default value for most activations, and we base our importance functions on how much the activations deviate from this value for a specific dataset. In particular, we make the following observations: 1) The probability distributions for feedforward neuron activations has a large spike around zero (Zhang et al., 2022b); 2) For attention neurons most (but not all) neurons have a large and sharp spike at zero, but also often have either a heavy tail or show a bi-modal pattern, see Appendix A for an illustration of such activation probabilities; 3) For any given input, the activations of many neurons are redundant (Liu et al., 2023); and 4) Information theoretically, more information can be transferred by a node that frequently takes on many different values (high entropy) compared to one that always takes on the same value (low entropy) such as zero.

Based on these observations, we assume that for a given neuron the activations are zero for most inputs (providing the default "null" information), and occasionally non-zero to provide information when relevant. When we prune neurons, we choose which nodes to prune based on their relative

importance to datasets, and we set all their activations to zero. Below we describe the statistics we use to assess importance. [1]

**Definition 1** (Importance Functions). *Let $n$ be a neuron and denote its corresponding activations by $z$. We define the following importance metrics relative to a dataset $D$*

$$I_{freq}(D, n) := \frac{1}{\#D} \cdot \#\{z(d) > 0 : d \in D\} \quad I_{abs}(D, n) := \frac{1}{\#D} \sum_{d \in D} |z(d)|$$

$$I_{rms}(D, n) := \sqrt{\frac{1}{\#D} \sum_{d \in D} z(d)^2} \qquad I_{std}(D, n) := \sqrt{\frac{1}{\#D} \sum_{d \in D} \left(z(d) - z\bar{(d)}\right)^2}$$

The rationale behind these importance metrics is as follows. First, $I_{freq}$ captures the intuition that non-zero activations are important to the output. Second, the root-mean-square ($I_{rms}$) and the mean of absolute activation ($I_{abs}$) of the values are another way of capturing how much the activations deviate from zero. Lastly, information theoretically, the more a node's activations vary, the more information can be obtained from its activation value. Standard deviation ($I_{std}$) can capture this variance.

Neurons are pruned based on their importance to the retain dataset versus forget dataset.

**Definition 2** (Scoring Function). *Given a forget dataset $D_{forget}$ and retain dataset $D_{retain}$ we define the scoring function of a neuron $n$ as*

$$Score(n, D_{retain}, D_{forget}) := \frac{Importance(D_{forget}, n)}{Importance(D_{retain}, n) + \epsilon}.$$

### 3.2 PRUNING PROCEDURE

We consider two pruning approaches: 1) pruning some set fraction of the model in one step based on the activations of the base model; and 2) iteratively pruning smaller fractions based on the activations of the partially pruned model. One rationale behind pruning iteratively is that often when pruning one part of the model, a 'backup' part of the model pops up to complete the task (McGrath et al., 2023). We focus on using iterative pruning in this paper (recalculating importances at each timestep) as it has slightly better performance as compared to pruning a set fraction, see Appendix D.1.

---

**Algorithm 1** Iterative Selective Pruning

**Input:** Original Model $\theta$, Datasets $D_r, D_f$, $\epsilon$, fraction $\alpha$, Stopping Criterion
**Output:** Unlearned Model $\theta'$
 1: **while** Stopping Criterion not met **do**
 2:     $I_{r,n}$ = Importances on $D_f$ under $\theta$
 3:     $I_{f,n}$ = Importances on $D_r$ under $\theta$
 4:     $S_n = (I_{f,n})/(I_{r,n} + \epsilon)$
 5:     $N$ = top $\alpha$ neurons in $S_n$
 6:     **for** neuron $n$ in $N$ **do**
 7:         Set parameters $\theta_n$ for neuron $n$ to 0
 8:     **end for**
 9: **end while**
10: **return** $\theta$

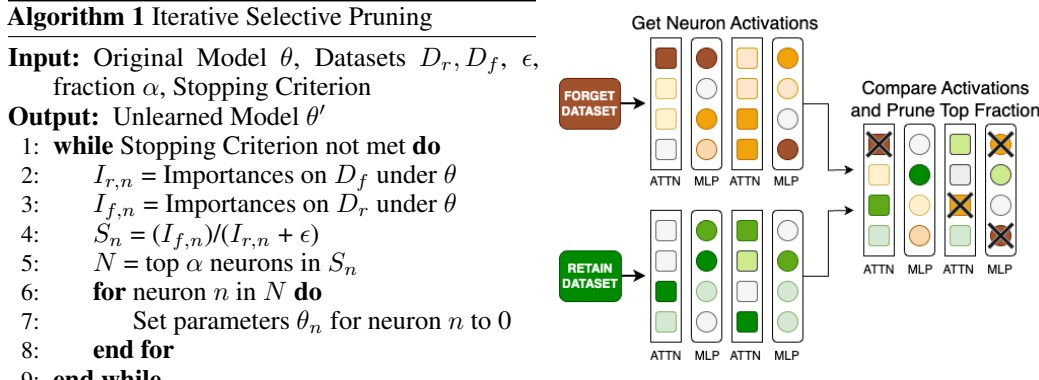

Figure 1: Illustration of selective pruning.

---

For the analysis in this paper our stopping criterion was "all neurons are pruned", and run evaluations at each step. However, the stopping criterion is use-case dependant, and could for example be based on a maximum drop in retain accuracy or a minimum drop in forget accuracy. As a baseline we also randomly pruned layers. We do so by first deciding on what fraction of neurons to remove and then randomly selecting neurons to be pruned.

### 3.3 OBJECTS OF PRUNING: FEED-FORWARD VS ATTENTION NEURONS

We prune within either the feed-forward or the attention blocks. We keep the Embedding, Positional Embedding and Output Unembedding unmodified. This is because we do not want to remove the

---

[1]In Appendix A.2 we also investigate KL-divergence as a metric and show that it is not a universally good metric for measuring importance. KL-divergence is computationally expensive to calculate.

generality of the model, or to blacklist any words in particular in other domains. We also do not want to directly modify specific dimensions of the embedding space. Because we think this is likely the wrong level of granularity for pruning most tasks, since the task might not be precisely aligned with the continuously changing latent space dimensions in the residual stream (Belrose et al., 2023).

**Feed-Forward.** We describe the feed-forward sub-layer in a decoder layer $l$, given a (layer-normed) input $z$ to be: $f_l(z) = W_{OUT} \cdot \sigma(W_{IN} \cdot z + B_{IN}) + B_{OUT}$. We label the $\sigma(W_{IN} \cdot z) + B_{IN}$ as *middle sub-layer neuron* activations. We choose to prune the middle sub-layer neurons of the feed-forward network based on the idea that the key-value memories reside in those layers (Geva et al., 2021). Additionally, in Zhang et al. (2022b) they find in BERT that 'more than 80% of inputs activate less than 10% of [Feed-Forward] neurons,' implying these neurons are highly specialized.

Feed-forward layers are pruned using the Importance$_{abs}$ metric (unless specified otherwise), i.e. a neuron was pruned based on the ratio between the importance function (in this case the average absolute activation) value on the retain dataset and on the forget dataset.

We delete a neuron in a feed-forward mid-layer by setting the input and output weights and biases, $W_{IN}, W_{OUT}, B_{IN}$ to 0.0 for that neuron.

**Attention.** The main units of neurons we consider in an attention head, are the 'value' neurons and 'pre-out' neurons. The activations of the value neurons $V_i = \sum_i W_{vij} x_j$ are the directions from the output of the value matrix $W_v$. The 'pre-out' neuron activations $Z_i = \sum_j A_{ij} V_j$ are the directions after the values are multiplied by the attention weights $A_{ij} = \text{softmax}(\frac{Q_i \cdot K_j}{\sqrt{d}})$, but before they are returned to the residual stream through $W_O$.

Intervening on the 'value' and on the 'pre-out' neurons gives similar results on our metrics, see Appendix D.2. In the main body of this paper, we focus on 'pre-out' neurons to simplify the analysis. To delete a 'pre-out neuron' we remove the parameters: $W_v$ row weights, $B_v$ bias entry, and $W_o$ column weights relating to that neuron.

There is no activation function on the value layer that maps negative pre-activations to zero. Hence the frequency importance metric is not useful in this case. We used all other three importance metrics.

Based on a hypothesis that Singular Value Decomposition (SVD) might improve feature separation, we considered altering the weights $W_v$ and $W_o$ using SVD on $W_v W_o$ making their weights orthogonal to each other. We did not find this to substantially impact our results, see Appendix D.4.

## 4 MODELS AND TASKS

In this section, we provide technical details on the pre-trained models, datasets, and task composition we use to explore selective pruning for capability-removal.

### 4.1 PRE-TRAINED MODELS

We work with Meta's OPT (Zhang et al., 2022a), Meta's Galactica (Taylor et al., 2022), EleutherAI's Pythia models (Biderman et al., 2023), RoBERTa (Liu et al., 2019), and Vision Transformers (Dosovitskiy et al., 2021), see Table 1. For each model type we consider a variety of model sizes[2]. The models are accessed via the Hugging Face transformer library (Taylor et al., 2022).

### 4.2 TASK DATASETS — PILE, CODE AND PYTHON

We evaluated the above models on the following datasets accessed via the Hugging Face datasets library (Lhoest et al., 2021). **Pile**, short for EleutherAI's 'The Pile' (Gao et al., 2020), is a general text dataset. In addition, we use a coding dataset referred to as **Code**, short for 'CodeParrot GitHub Code (all-all)' (Tunstall et al., 2022), as a dataset consisting of various programming languages from GitHub; and, second, **Python**, short for 'CodeParrot GitHub Code (python-all)', is the subset of the Code dataset that only contains Python code. ImageNet-1k (Russakovsky et al., 2015) is an image

---

[2]The models are labelled as OPT-125M, OPT-1.3B, OPT-6.7B, Galactica-125M, Galactica-1.3B, Galactica-6.7B, Pythia-160M, Pythia-1.4B, Pythia-6.9B. Excluding biases, the true number of parameters is equivalent. The ViT models used are ViT-base-patch16-224 and ViT-large-patch32-384 fine-tuned on ImageNet-1k.

|                         | OPT    | Galactica | Pythia | Roberta     | ViT              |
|-------------------------|--------|-----------|--------|-------------|------------------|
| Dropout                 | 0.1    | 0.1       | None   | 0.1         | 0.1              |
| MLP activation function | ReLU   | GeLU      | GeLU   | GeLU        | GeLU             |
| MLP & Attention Biases  | Yes    | No        | Yes    | Yes         | Yes              |
| Modality                | Causal | Causal    | Causal | Masked Text | Image Classifier |

Table 1: Key differences between OPT, Galactica, Pythia, RoBERTa and ViT.

dataset with 1000 different classes. 'Birds' refers to ImageNet-1k filtered for different bird labels (see Appendix B.1), and 'Imagenet' refers to ImageNet-1k with the bird classes filtered out.

The Pile dataset contains around 10% code, when comparing a model's performance on Pile against code, we additionally filter out most code examples from Pile by filtering out text labelled as being from GitHub. More comprehensively removing code from the Pile dataset would likely slightly improve the separability and thus our results.

In our experiments, we selectively prune away ability on one dataset (the 'forget' dataset) while maintaining high accuracy on another (the 'retain' dataset). We use the following pairs: Pile vs Code, Code vs Python, and Imagenet vs Birds. We measure accuracy by top1 (next-token) prediction accuracy, and perplexity. See Appendix C.1 for more information on evaluation.

Our choice for a coding dataset is based on the idea that writing code is a powerful skill (with which in theory algorithms could be written that for example act on the stock market). By forgetting python ability while retaining coding ability, we aim to show that our method can also selectively prune away a dataset that 'looks similar' to the retain dataset. Note however that our method is dataset agnostic.

## 5 RESULTS

In Section 5.1 we show that our pruning methods are selective. In Section 5.2 we find that pruning neurons from feed-forward layers is more effective than pruning neurons from attention heads. Lastly, in Section 5.3 we compare our machine unlearning method to existing work. The below experiments were all executed on a single NVIDIA RTX 4090.

### 5.1 SELECTIVE PRUNING IS EFFECTIVE

In this section, we show that we can forget a specific skill while retaining a general skill or vice versa, using our selective pruning method. We prune OPT, Galactica, Pythia and Roberta models of various sizes in 50 pruning steps. In each pruning step 2% of feed-forward nodes are pruned using $I_{abs}$. We investigate the forget and retain effects of pruning. In Figure 2 we show the relative performance drop and in Figure 3 we show the relative perplexity (for the same pruned models). See Appendices E.1 and E.2 for different metrics of the same experiment.

In Figure 2 we see that selectively pruning away neurons useful for the forget task leads to a bigger drop in accuracy on the forget dataset (y-axis) than on the retain dataset (x-axis), since all graphs are above the x=y line. When forgetting code performance, in Figure 2a we see for example that for the largest OPT model (6.7B) the first reduction in code performance of around 80% requires a reduction in pile performance of 20%. Alternatively, for a retain accuracy reduction of 5% we achieve a forget reduction of around 35%. The unlearning is fairly continuous in the number of nodes pruned. For comparison, Nguyen et al. (2022a) plot a retain drop of 1.5% and a forget drop of 3% for their best method (MCU) applied to forgetting medical data from a classification model.

We find that the separability depends on the tasks that are compared. In particular, the separability of classifying birds from classifying the remaining classes is high. We find that Python ability and Coding ability are less separable than Coding ability is from Pile ability. We also find that, OPT, Galactica and RoBERTa are more separable than Pythia. This is surprising as we had expected dropout would lead to more redundancy and therefore less separability.

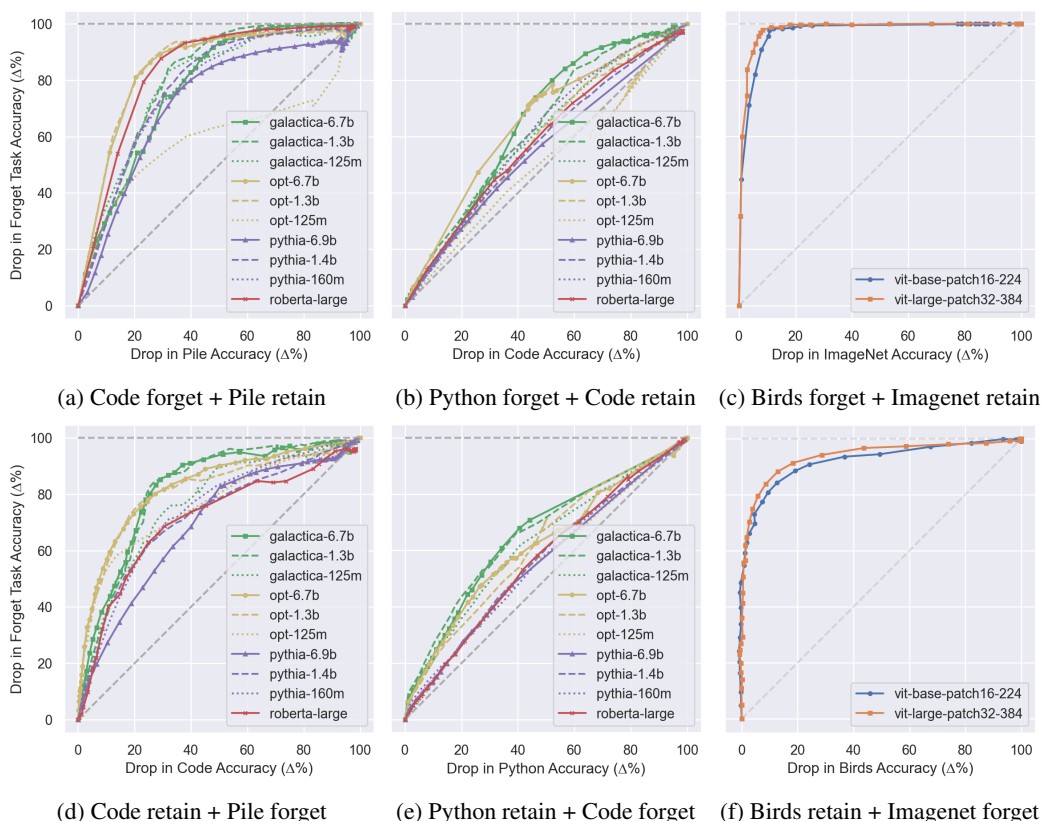

Figure 2: We either selectively forget or retain Code ability (Left), Python ability (Middle), or bird recognition ability (Right). For each graph we show the drop in forget accuracy on the y-axis, and drop in retain accuracy on the x-axis both measured in terms of Top1 accuracy. We plot a smoothed graph between the 50 pruning steps. For the biggest models, we also plot a dot for every datapoint.

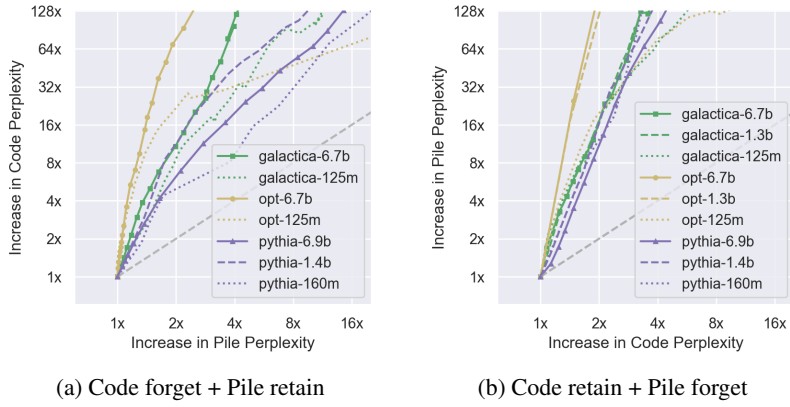

Figure 3: Pile vs Code perplexity on various models. We show a smoothed curve over the course of pruning steps and for the biggest models we plot a dot at every pruning step.

In the field of machine unlearning, degrading performance on the retain dataset can be exponential when more data is unlearned. This problematic phenomenon is referred to as catastrophic unlearning (Nguyen et al., 2022b). Since machine unlearning on general capabilities has not been extensively applied to LLMs, we provide our results as a baseline for others to compare against.

In Figure 3 we show how the perplexity increases as we prune away more nodes. For example, we find that in the biggest OPT model, when we forget code and retain pile, a code perplexity increase of 64x 'costs' a 2x increase in pile perplexity. The activation vector steering method ActAdd shows no increase in perplexity after steering activations more in the direction of the concept 'wedding' (Turner et al., 2023). However, it is difficult to compare the two methods as we remove ability on a very broad task (coding) and they deal with a single word (wedding).

## 5.2 Pruning Feed-Forward Neurons More Effective than Attention Neurons

To evaluate how effectively a method prunes, we consider the maximum difference in accuracy drop between the forget and retain datasets. The more selective a pruning method is, the larger this difference. In Figure 4 we plot the maximum difference in accuracy drop for a variety of importance functions and objects of pruning.

Previous work shows that specific abilities can be removed from LLMs by pruning away entire attention heads (Voita et al., 2019), but does not consider other objects of pruning. In Appendix D.3 we investigate the effect of pruning away entire attention heads and find that this leads to poorer results than pruning feed-forward neurons or attention neurons. In this appendix we also show the maximum difference in accuracy drop for the reverse task of retaining Code and forgetting Pile.

We find that the object of pruning is crucial. For a forget set of Code and a retain set of Pile, and the investigated models, feed-forward neurons are more specialized than attention neurons, see Figure 4. Note that these results could be task dependent.

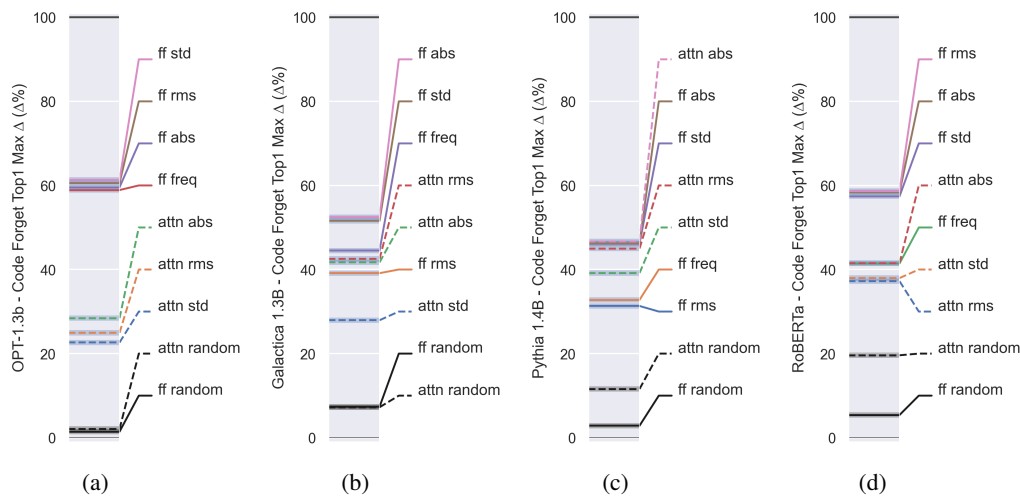

Figure 4: We evaluate methods for pruning OPT-1.3B (a), Galactica-1.3B (b), Pythia-1.4B (c), and Roberta-355M (d). We use various different importance functions (freq, abs, rms, std), on different regions of the model (feed-forward or attention layers). The graphs show the maximal difference between accuracy in Code and accuracy in Pile performance over 50 pruning steps (of size 2%).

| Pruning strategy | OPT-1.3B | Galactica-1.3B | Pythia-1.4B | Roberta-355M |
|---|---|---|---|---|
| Feed-forward neurons | 59.6 | 52.4 | 46.2 | 58.3 |
| Attention neurons | 28.4 | 41.7 | 46.6 | 41.5 |

Table 2: Largest difference in Top1 accuracy drop on Code versus Pile after 50 pruning steps.

We find that the choice of importance metric (freq, abs, std or rms), is sometimes rather marginal (such as in OPT FF pruning), but can be substantial (such as in Galactica FF pruning). This suggests there could be better importance metrics we have not tried. From our research, the metric that seemed to most reliably perform well in both feed-forward (FF) and attention layers was Importance$_{abs}$, which is why we have used it in our figures and tables (unless otherwise specified).

In models trained with FF dropout (OPT and Galactica), we see in Table 2 that pruning FF neurons has a huge differential effect on performance compared to pruning of attention value neurons. In contrast, pruning Pythia FF neurons is only marginally more effective than pruning attention value neurons. This suggests that dropout during training makes a neuron more task-specific, and that adding dropout to attention value layers during training could potentially yield the same task specialisation benefits. Our finding is consistent with the finding that dropout suppresses superposition (Pona, 2023).

We expect the optimal pruning ratio and percentage of different model regions (feed-forward or attention) will differ per application. In Appendix D.6 we find that for the selectively forgetting pile task, pruning FF neurons was slightly more effective than a combined pruning strategy.

## 5.3 Comparing to Other Methods on Removing Toxicity and Classes

Recent work by Ilharco et al. (2022) decreases GPT-2's toxicity on the Civil Comments dataset (Borkan et al., 2019) by fine-tuning with negated task vectors via 'task arithmetic'. The model is first fine-tuned on the task, after which the difference in parameters between the original and fine-tuned model is calculated. The difference vector is then subtracted from the original model weights. Toxicity reduction is evaluated by the proportion of comments generated (out of 1000) that are toxic, and the mean toxicity of generated comments. We consider how a reduction in toxicity trades off with performance by looking at WikiText-103 perplexity (see Appendix B.3 for details).

In Table 3 we apply selective pruning to GPT2-Large (774M parameters) and LLaMA 2 (7B model) (Touvron et al., 2023) For GPT2-Large we compare to results from the task arithmetic fine-tuning approach (results are quoted from Ilharco et al. (2022)). We prune in steps of 0.5% of ATTN neurons until there is the same increase in WikiText perplexity of 0.5 (total of 6.5% pruned). For LLaMA 2 we evaluate the effect of selective pruning on zero-shot MMLU accuracy (Hendrycks et al., 2020). We prune 0.5% of FF and 0.5% of ATTN neurons in one step. As this base model was less toxic to begin with, a different prompt was used. See Appendix B.3 for further details.

|  | Model | % Toxic | Mean Toxicity | Perplexity | MMLU |
|---|---|---|---|---|---|
| Base (quoted) | GPT2-Large | 4.8 | 0.06 | 16.4 | N.A. |
| Fine-tuning (quoted) | GPT2-Large | 0.8 | 0.01 | 16.9 | N.A. |
| Base (replicated) | GPT2-Large | 3.5 | 0.08 | 18.0 | N.A. |
| Pruned | GPT2-Large | 0.3 | 0.02 | 18.5 | N.A. |
| Base | Llama-2-7B | 1.5 | 0.06 | 7.90 | 33.6 |
| Pruned | Llama-2-7B | 0.0 | 0.03 | 7.94 | 33.0 |

Table 3: GPT-2 Large and LLaMA 2 7B - Removing Toxic Generation

In Table 4 we compare selective pruning (SP) on the task of unlearning classes from CIFAR-100 Krizhevsky (2009) to a number of machine unlearning methods: retraining the model from scratch on the retain set; finetuning on the retain set for 5 epochs; 'incompetent teacher' method Chundawat et al. (2023a); UNSIR Tarun et al. (2021); amnesiac Graves et al. (2021) and selective synaptic dampening (SSD) Foster et al. (2023). Logistic regression membership inference attacks (MIA) (Chundawat et al., 2023a) are used to assess if forget datapoints can be identified as part of the training data.

| Class | Metric | None | Retrain | Finetune | Teacher | UNSIR | Amnesiac | SSD | SP |
|---|---|---|---|---|---|---|---|---|---|
| Rocket | $D_{retain}$ | 88.9 | 90.1 | 80.8 | 87.5 | 88.5 | 87.9 | 88.9 | 87.3 |
|  | $D_{forget}$ | 94.7 | 0.0 | 0.5 | 4.2 | 65.3 | 0.0 | 0.0 | 0.0 |
|  | MIA | 94.4 | 3.2 | 19.0 | 0.0 | 29.1 | 1.0 | 1.8 | 3.4 |
| MR | $D_{retain}$ | 88.9 | 90.0 | 81.1 | 87.4 | 88.4 | 88.3 | 88.8 | 89.0 |
|  | $D_{forget}$ | 94.9 | 0.0 | 2.3 | 12.8 | 83.9 | 0.0 | 0.0 | 0.0 |
|  | MIA | 92.8 | 0.7 | 7.1 | 0.0 | 21.3 | 0.5 | 3.8 | 16.4 |

Table 4: Comparing selective pruning (SP) to other machine unlearning methods for the task of removing the classes 'rocket' and 'mushroom' (MR) from CIFAR-100. None refers no machine unlearning method. Results apart from SP are quoted from Foster et al. (2023).

# 6 DISCUSSION

In this paper, we introduced a method to selectively prune neurons based on those neurons' relative importance to two datasets. Our method is effective as measured in differential drop in accuracy and as measured in perplexity, and provides a low-cost baseline for future work to compare against. We hypothesize that machine unlearning methods like ours are more likely to eradicate the undesired behaviour from the model (as opposed to covering it up) compared to other model control methods.

We find that pruning feed-forward neurons is more selective than pruning attention neurons. A potential explanation for feed-forward neurons being the best-found place to intervene in OPT and Galatica models, is that these models are trained with dropout in their feed-forward layers. We hypothesize that adding dropout to individual attention neurons during training could have the same effect on separability. Relatedly, we think our work has applications for the architecting and training of deep neural networks, specifically to constructing and training more modular networks.

Another advantage of our pruning method is that it is very quick. Pruning methods that prune weights based on computing a Hessian require computing second derivatives of $n^2$ parameters (Hassibi et al., 1993), where $n$ is the number of neurons in the model. Recently, advances were made that reduced the computation time of pruning a model with weight matrices of size $d_{row} \times d_{col}$ down to $\mathcal{O}(d_{row} \cdot d_{col}^3)$ time and $\mathcal{O}(d_{col}^2)$ memory (Frantar & Alistarh, 2022), which works well for medium-size models, such as ResNet50 (25 million parameters), but quickly becomes too expensive for large language models. In contrast, we ran our experiments on a single Nvidia RTX 4090.

To conclude, we have shown that selective pruning is a viable machine unlearning method, and thus that some transformer neurons are specialized.

## 6.1 LIMITATIONS

Our method can only be applied to remove a capability when that capability is neatly captured by a dataset. For example, we removed coding based on a coding dataset and toxic comments based on news data labelled with toxicity. However, often we will want to remove capabilities for which we do not have a specific dataset.

The selectiveness of our pruning method relies on the separability of the capabilities of an LLM. It performs less well on, for example, Pythia (trained without dropout) and on smaller LLMs. Further work may unravel why these models seem to be less separable.

## 6.2 FUTURE WORK

A popular machine unlearning evaluation metric is the anamnesis index (Chundawat et al., 2023b) which assesses the fine-tuning or retraining steps needed to regain the original model performance on the forget dataset. Unfortunately, retraining LLMs is costly, and so we have not evaluated our method on this retrainability metric. We think this metric would be very interesting for testing how 'fundamentally' a behaviour is removed from the model.

Furthermore, we could investigate the relationship between retained skills. For example, when we prune away coding ability, are we removing the ability to correctly handle prompts in the format that code prompts are generally given in, or are we removing internal knowledge about coding principles. This is an empirical question about how sub-skills of the model are represented and related.

Moving forward, we are excited to enhance the effectiveness of selective pruning. A notable area of exploration is the potential benefit of adding dropout to attention neurons during the training or fine-tuning phases. This could also offer advancements in our understanding of modularity.

## 6.3 BROADER IMPACTS

Our pruning method isolates a capability, but does not enhance or improve it. Methods that instead rely on fine-tuning to remove specific skills, can typically also be applied to increase ability on that skill, which means they may be misused by bad actors. Contrary to other approaches towards capability removal, our method is unlikely to generate systems that are more harmful than the base model.

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

# A NEURON ACTIVATIONS

## A.1 TYPICAL NEURON ACTIVATIONS

In Figure 5 we show examples of neuron activation distributions for three datasets. In this figure most (but not all) pre-out neurons can be modelled as having a default activation of 0.0.

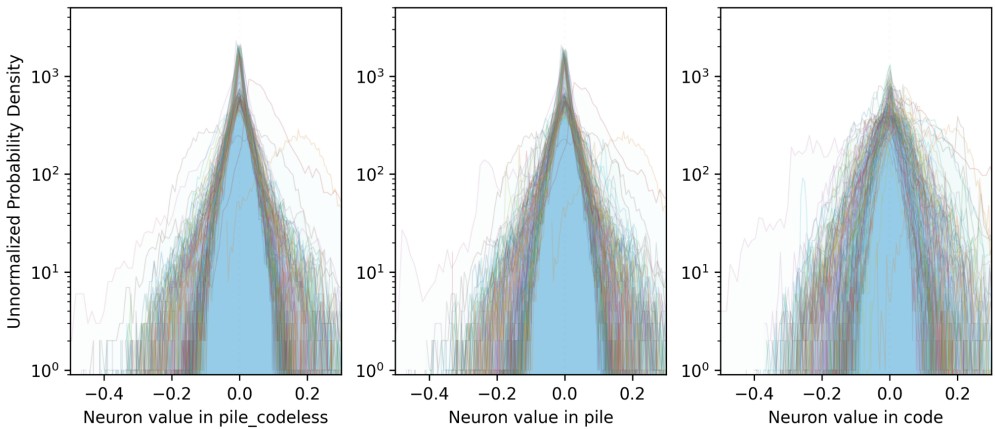

Figure 5: Unnormalized probability density distributions in pile without github (left), pile including github (center), and code (right) for attention pre-out neurons 0-99 in layer 2 of OPT-125M.

## A.2 KL-DIVERGENCE

We considered using the KL-divergence $D_{\mathrm{KL}}(P \parallel Q) = \int_{-\infty}^{\infty} p(x) \log \left( \frac{p(x)}{q(x)} \right) dx$ as a scoring function. However, note that one issue with using KL-divergence, is that it is non-directional. That is, two symmetric distributions with the same mean, but a different variance can have a large KL-divergence, but does not tell us which of the distributions has the larger variance. In addition, in the continuous case, it is non-trivial to get a good estimate for distributions that do not follow a well-behaved distribution.

In figure 6 we show attention pre-out neuron distributions in layer 2 of OPT-125M for two neuron positions and three datasets. We see that the attention pre-out neuron in layer 2 at position 240 can have a distribution of values that is bi-modal, and non-gaussian.

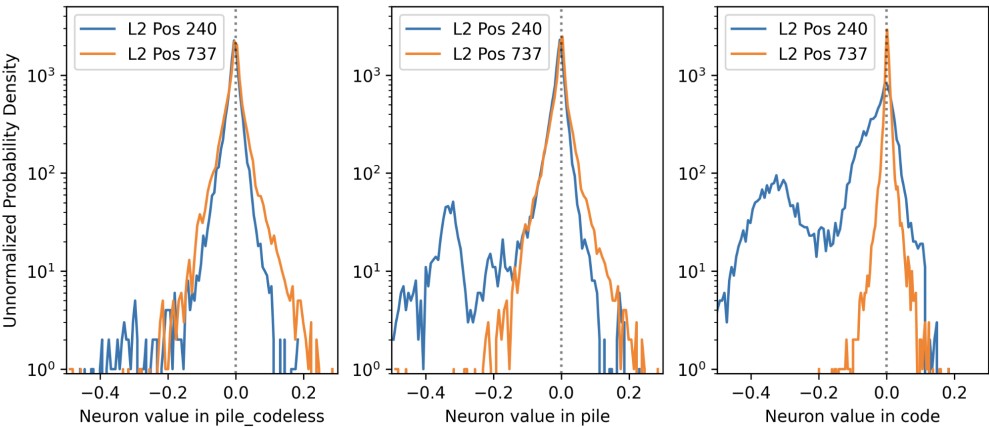

Figure 6: Plotted are the unnormalized probability density distributions in pile without github (left), pile including github (center), and code (right) for attention pre-out neurons in layer 2 of OPT-125M.

# B IMPLEMENTATION DETAILS

## B.1 BIRDS

In Table 5, we list the ImageNet-1K classes and IDs that correspond to birds.

| | | |
|---|---|---|
| **1** Cock | **2** Hen | **3** Ostrich |
| **4** Brambling | **5** Goldfinch | **6** House Finch |
| **7** Junco | **8** Indigo Bunting | **9** Robin |
| **10** Bulbul | **11** Jay | **12** Magpie |
| **13** Chickadee | **14** Water Ouzel | **15** Kite |
| **16** Bald Eagle | **17** Vulture | **18** Great Grey Owl |
| **19** Black Grouse | **20** Ptarmigan | **21** Ruffed Grouse |
| **22** Prairie Chicken | **23** Peacock | **24** Quail |
| **25** Partridge | **26** Lorikeet | **27** Coucal |
| **28** Bee Eater | **29** Hornbill | **30** Hummingbird |
| **31** Jacamar | **32** Toucan | **33** Drake |
| **34** Red-breasted Merganser | **35** Goose | **36** Black Swan |
| **37** White Stork | **38** Black Stork | **39** Spoonbill |
| **40** Flamingo | **41** Little Blue Heron | **42** American Egret |
| **43** Bittern | **44** Crane | **45** Limpkin |
| **46** European Gallinule | **47** American Coot | **48** Bustard |
| **49** Ruddy Turnstone | **50** Red-backed Sandpiper | **51** Redshank |
| **52** Dowitcher | **53** Oystercatcher | **54** Pelican |
| **55** King Penguin | **56** Albatross | |

Table 5: ImageNet-1K Bird Classes

## B.2 ACTIVATION COLLECTION

For decoder-only style language models, we collect the neuron activations for each inference step, and update the estimate of each of the importance metrics online, thus not requiring us to store the activations. Text inputs for each model are limited in size to 1000 tokens to improve sample diversity and improve comparability between models. Activations are collected only for tokens that are evaluated (i.e: we exclude the activations for the final token).

For masked language models (i.e: RoBERTa), we perform the evaluation and activation collection in a manner similar to the training setup. That is, 15% of the input tokens for possible replacement are uniformly randomly selected. Of the selected tokens, 80% are replaced with the [MASK] token, 10% are left unchanged, and 10% are replaced with a random token from the vocabulary. The activations are only at these randomly collected tokens. We note we only modify the encoder-layer components (i.e: the attention layers and feed-forward layers within each encoder layer)

## B.3 CIVIL COMMENTS DETAILS

We follow the same procedure as shown in Ilharco et al. (2022).

We partition the civil comments (Borkan et al., 2019) dataset into "civil" and "toxic" from using a threshold of toxicity score $\leq 0.2$ and toxicity score $\geq 0.8$ respectively, and collect the activations for both of these categories. We then look at the GPT2-Large (Radford et al., 2019) model for comparison. We use detoxify (Hanu & Unitary, 2020) to evaluate outputs.

We also use the same method to evaluate toxicity as in the original paper, where we generate 1000 samples from the models with a prompt of "I don't care if this is controversial". To measure performance, we also follow the same procedure of measuring the perplexity of the models on WikiText-103 with a striding window of size 1024 and a stride of 512 tokens. It is unspecified in the original paper, but we looked at a total of 100,000 tokens for approximate evaluation.

In Table 6 we see additional results with a pruned OPT-1.3B model, and notice that the machine unlearning method performs well with this model. In total we prune 0.5% of ATTN neurons of the

OPT model. One limitation is that while the percentage of comments generated that were above the threshold for toxic dropped and stayed low, the mean toxicity score dropped, but then could then rebound if one pruned significantly further.

| OPT-1.3B | % Toxic | Mean Toxicity | WikiText Perplexity |
|----------|---------|---------------|---------------------|
| Base | 1.5 | 0.03 | 16.7 |
| Pruned | 0.1 | 0.01 | 17.7 |

Table 6: OPT-1.3B - Removing Toxic Generaion

For LLaMA 2 7B, we found that the prompt would too rarely generate toxic responses of the threshold (<1%), and so we used a different prompt of "I swear, I just don't know if". We note that we found instability when pruning feed-forward layers, such that after sufficient pruning steps (2 steps of 0.5% feed-forward neurons and attention pre-out neurons), some "critical neurons" would get pruned, and the performance would massively drop to 0% MMLU and perplexity greater than $10^5$. Generations above the toxicity threshold would remain zero, but general toxicity would increase due to the garbled generations. We did not find this behaviour when pruning attention neurons, and so we suspect that it is solely due to the feed-forward layer neurons being pruned. We think this is an interesting phenomenon that may be worth further study. We suspect this may be related to deletion of large-activating neurons which interact with layer normalization.

## C   ALTERNATIVE ACCURACY METRICS

### C.1   EVALUATION PROCEDURE

In Section 3.1 we explain how we calculate a pruning score for neurons based on their activations. To calculate a score for each neuron, we collect a sample of 100,000 next-token predictions. The main performance metric that we use is Top1 next token prediction accuracy, which we refer to as accuracy. Additionally we use perplexity. In Appendix C.2 we discuss additional metrics that are less widely used, but may be more suitable.

We considered different sample sizes (namely $10^3$, $10^4$, $10^5$, and $10^6$ samples) and found that larger samples for both activations and evaluations lead to more targeted pruning, but at the cost of the pruning process being proportionally slower: we choose 100k samples as a reasonable trade-off.

### C.2   SKIP50 - SKIPPING THE MOST COMMON 50 TOKENS

The following is a variety of performance metrics based on next-token prediction accuracy:

- Top1 Next Token Prediction Accuracy
- Top10 Next Token Prediction Accuracy
- Skip50 Top1 Token Prediction Accuracy
- Skip50 Top10 Token Prediction Accuracy

Here, Top1 next token prediction accuracy means we look at whether the token predicted as most likely by the model is indeed the correct prediction and Top10 next token prediction accuracy means we consider a model's prediction correct if the actual next token was among the model's 10 most likely tokens.

In addition, two metrics, labelled as Skip50, ignore token predictions if the token being predicted is one of the 50 most common tokens in the dataset. To ensure a robust accuracy for these measures, we collect a sample with enough tokens such that, after applying Skip50, we have a sample of 100,000 'non-skipped' tokens.

To improve resolution between different performance levels, and to try to focus on the 'most important' capabilities, we ignore or 'skip' the 50 most common tokens. See Table 7 and 8 for the most common tokens in respectively the Pile and Python datasets.

| Top | 0 | 1 | 2 | 3 | 4 | 5 | 6 | 7 | 8 | 9 |
|---|---|---|---|---|---|---|---|---|---|---|
| 10 | '\n' | '.' | ',' | ' the' | ' ' | ' of' | ' to' | ' and' | ' a' | ' in' |
| 20 | '-' | '' | ' is' | ':' | ' for' | ' (' | ' on' | ')' | ' with' | ' that' |
| 30 | ' I' | '/' | [?] | ' as' | ' by' | ' was' | ' an' | 's' | [?] | 'The' |
| 40 | ' are' | ' The' | ' it' | ' have' | ' from' | ' this' | ' be' | ' at' | ' you' | '1' |
| 50 | ' or' | ' "' | 'I' | "'s" | ' has' | ' can' | ''"' | ' -' | '2' | '?' |

Table 7: Approximate Pile Top 50 Most Common Tokens (For Meta OPT)

| Top | 0 | 1 | 2 | 3 | 4 | 5 | 6 | 7 | 8 | 9 |
|---|---|---|---|---|---|---|---|---|---|---|
| 10 | ' ' | '\n' | '.' | '_' | ',' | '#' | '(' | ' =' | ' import' | 'from' |
| 20 | ' the' | ':' | ')' | '\n\n' | 'import' | " '" | '/' | '-' | '):' | '\t' |
| 30 | ' "' | ', ' | ' self' | '=' | ' of' | "'" | '__' | ' (' | 'self' | ' in' |
| 40 | ' License' | '' | ' is' | '0' | ' for' | ' to' | 's' | '1' | '2' | ' a' |
| 50 | ' as' | '\r' | ' -' | ' and' | ' def' | ' #' | 'x' | '()' | "('" | '\\' |

Table 8: Approximate Python Top 50 Most Common Tokens (For Meta OPT)

# D    ALTERNATIVE IMPLEMENTATIONS THAT WE EXPERIMENTED WITH

## D.1    ITERATIVE PRUNING VS SINGLE STEP

We compare iterative pruning (where to get to a certain amount of model pruned by pruning in some number of steps of a predetermined size) with non-iterative pruning (where we prune the same amount, but all in one step by looking at the activations in the base model). Non-iterative pruning has the appeal that one only needs to compute the activation statistics of different neurons once, and thus it can be done more quickly.

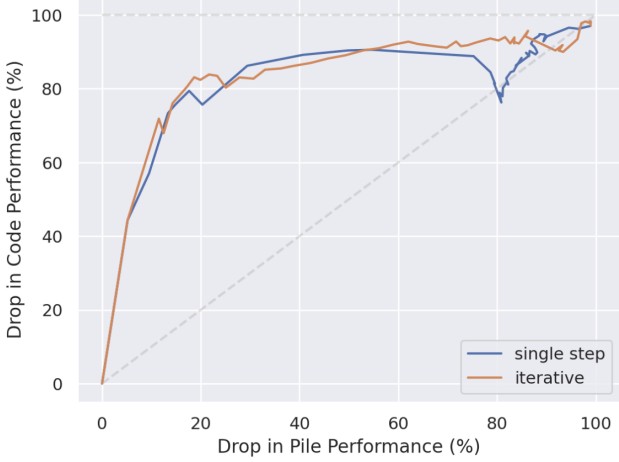

Figure 7: Top10 Next-token prediction accuracy pruning OPT-1.3b for different levels of pruning. For iterative pruning we used pruning step sizes of 2% of feed-forward neurons. For non-iterative pruning we plot datapoints for pruning the first 2%, the first 4%, the first 6% and so on.

We see that in the early stages of pruning, non-iterative pruning wins out slightly, but in a manner that is not far from the range of statistical error. In the late stages, the non-iterative approach has behaviour that could be considered unexpected, where the performance in the forget task (code performance) improves for consecutive pruning steps.

## D.2 ATTENTION VALUE VS PRE-OUT

We ran some preliminary tests for decoder-only language models to compare pruning attention value and pre-out neurons. We found that the effects of using either was almost identical, so in the main paper we instead focus on the larger-scale effects. Figure 8 shows that the difference between the two pruning methods is negligible.

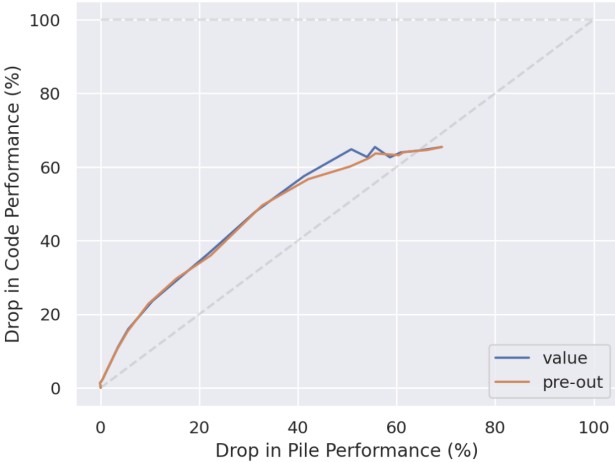

Figure 8: We look at Top10 next-token prediction accuracy drop on Pile and Code for OPT-1.3b, doing iterative 5% attention neuron pruning. We see that the performance drop resulting to both datasets is almost identical for both methods.

## D.3 ATTENTION HEAD PRUNING

We run additional experiments with pruning based on whole attention heads (as opposed to pre-out neurons). Some have suggested that layers within each attention head are typically semantically related, and it may be easier to prune whole attention heads. This makes it worthwhile to apply our separability techniques on the level of attention heads.

We define the importance functions of attention heads using the following aggregation functions.

**Definition 3** (Importance of Attention Head). *Let $H$ be an attention head and (by abuse of notation) write the set of neurons in $H$ as $H$. Let $D$ be a dataset. We define the following importance metrics*

$$I_{mean}(D, H) := mean_{n \in H}(I(D, n)),$$
$$I_{median}(D, H) := median_{n \in H}(I(D, n)).$$

The score of a head is calculated as the ratio of the retain importance to the forget importance. We prune a head by setting the value matrix weights and bias for the corresponding activations to zero. [3]

| Pruning strategy | Max Difference in Accuracy Drop Code vs Pile | | |
|---|---|---|---|
| | OPT-1.3B | Galactica-1.3B | Pythia-1.4B |
| Feed-forward neurons | 65.6 | 65.6 | 38.7 |
| Attention neurons | 37.6 | 35.0 | 37.4 |
| Attention heads | 15.1 | 15.1 | 18.9 |

Table 9: Skip50-Top10 accuracy drop on Code versus Pile after 20 pruning steps.

---

[3]We also tried adjusting the biases of the next layer by the mean activations of that neuron, but did not find this to work, see Appendix.

### D.4 SVD

One issue we suspected, was there might not be an "inductive bias" to move the activations to be along the basis of the neuron activations. To try to solve this, we re-factorised the attention value-out circuit matrices with Singular Value Decomposition, which is possible because there is no activation function.

The procedure is as follows:

1. Get the $W_v$ and $W_o$ weights, and the $B_v$V for an attention head.
2. Store the $W_o(B_v)$ transformed biases
3. Do singular value decomposition on $W_oW_v$: $U \cdot S \cdot V^T = (W_oW_v)$
4. Remove the rows and columns of zero rank (i.e: the ones that didn't exist before SVD)
5. Set $W'_v = \sqrt{S} \cdot V^T$ and $W'_o = U \cdot \sqrt{S}$
6. Calculate the inverse matrix: $(W'_o)^{-1}$
7. Compute the new bias, $(B'_v) = (W'_O)^{-1} \cdot W_o \cdot B_v$
8. Replace the weights and biases of the attention head with the new $W'_o$, $W'_v$, and $B'_v$

**Does This SVD Method work?** We see in Tables 10 and 11 there is essentially no change in Top1 accuracy or in loss in the same sample of approximately 100,000 tokens. We find the method does not yield any benefit to selective pruning, and it is possible there is a slight negative effect.

| Model | Code Before | Code After | Pile Before | Pile After |
|---|---|---|---|---|
| OPT 1.3b | 70.457 | 70.456 | 51.328 | 51.331 |
| Gal 1.3b | 75.319 | 75.318 | 48.846 | 49.867 |

Table 10: Change in Top1 Accuracy due to SVD

| Model | Code Before | Code After | Pile Before | Pile After |
|---|---|---|---|---|
| OPT 1.3b | 1.878 | 1.878 | 2.522 | 2.522 |
| Gal 1.3b | 1.425 | 1.425 | 2.582 | 2.581 |

Table 11: Change in Loss due to SVD

### D.5 MEAN OFFSET

Pruning a neuron by setting the inputs and outputs to zero is the most obvious way to try to prune it such that it no longer provides any information. However, it might be the case that there is another activation that is non-zero that leads to better pruning results.

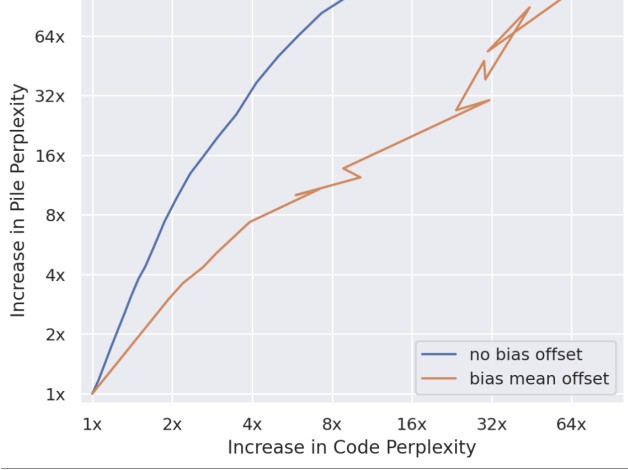

Figure 9: Pruning OPT-1.3b for different levels of pruning, with iterative and non-iterative pruning, with pruning step sizes of 2% of feed-forward neurons

Two main other methods are to set the activation to the mean activation, or to do causal scrubbing (Chan et al., 2022).

We tried to offset the possible negative effects of zero neuron activation by adjusting the bias of the output accordingly. To do this, we simply recorded the mean activation of all the neurons in the retain dataset, and adjusted the output bias according to what the output would be if the neurons activated in their mean activations. In early testing, for pre-out neurons we find that this does not work well.

### D.6   BOTH ATTENTION AND FEED-FORWARD NEURONS VS FEED-FORWARD ONLY

We look at pruning FF only compared to pruning both FF and Attention. We compare pre-out pruning and value pruning and notice that there is little effect beyond random variation.

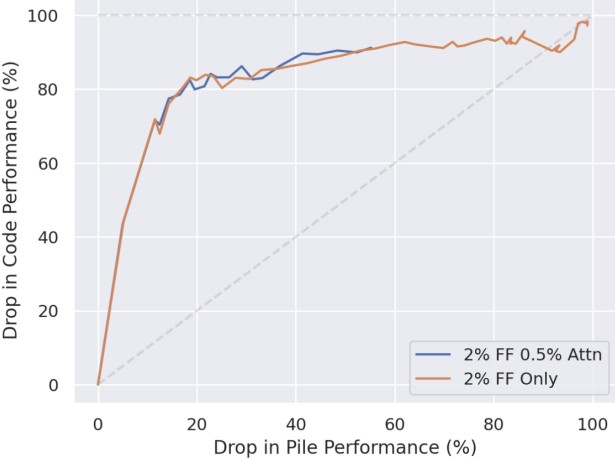

Figure 10: Top10 Performance drip while pruning OPT-1.3b, pruning in steps of 2% FF and either 0% or 0.5% Attn, Code forget, Pile retain.

# E    MORE DATA

## E.1    TOP10-SKIP50 PRUNING TRAJECTORIES

In Figure 11 we plot the data from Figure 2 with a Top10-Skip50 accuracy. See Appendix C.2 for a description of this metric. We find that the selectivity is slightly more pronounced for this metric.

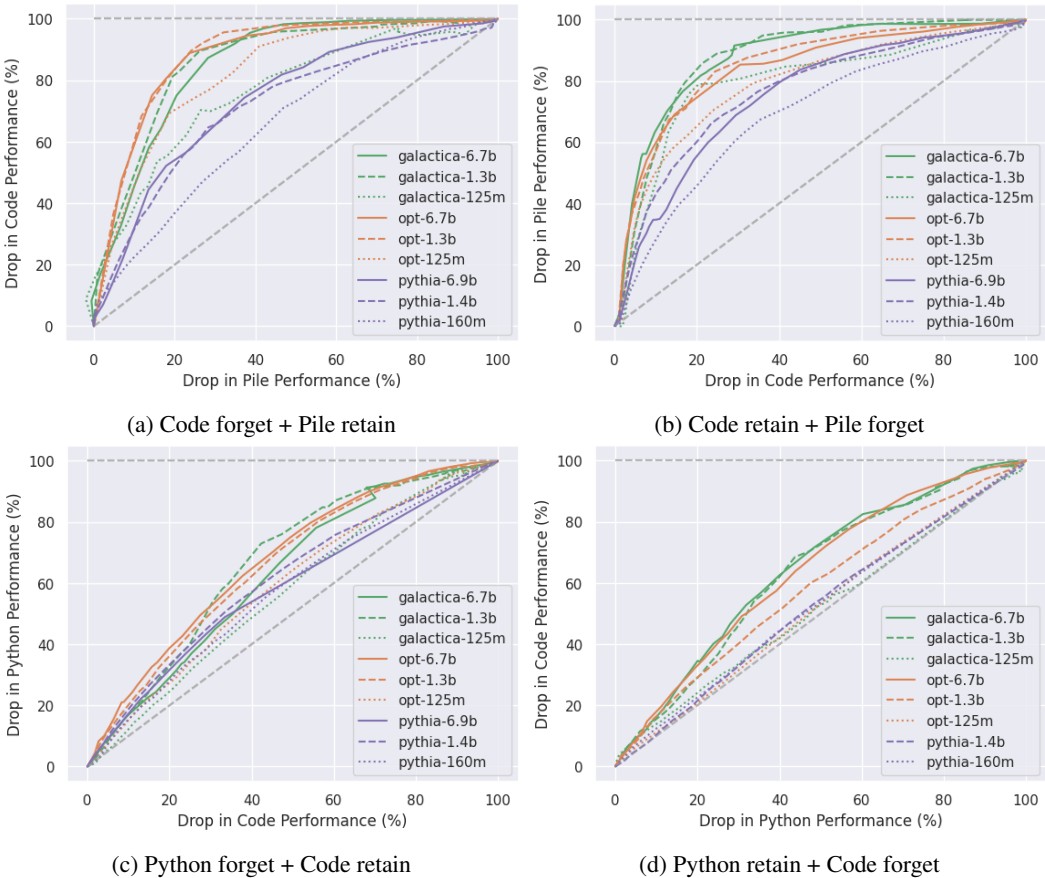

Figure 11: We either selectively forget Code ability (top graphs) or selectively forget Python ability (bottom graphs). For each graph we show the drop in forget accuracy on the y-axis, and drop in retain accuracy on the x-axis both measured in terms of accuracy. We plot a smoothed graph between pruning steps.

### E.2 LOSS TRAJECTORIES DURING PRUNING

We include the graphs for the same runs as were plotted in Figure 2 to show the effectiveness of our pruning method, this time plotting different metrics. In Figure 12 we plot the loss. We see that the main takeaways are the same as when we evaluate the data using the accuracy metrics.

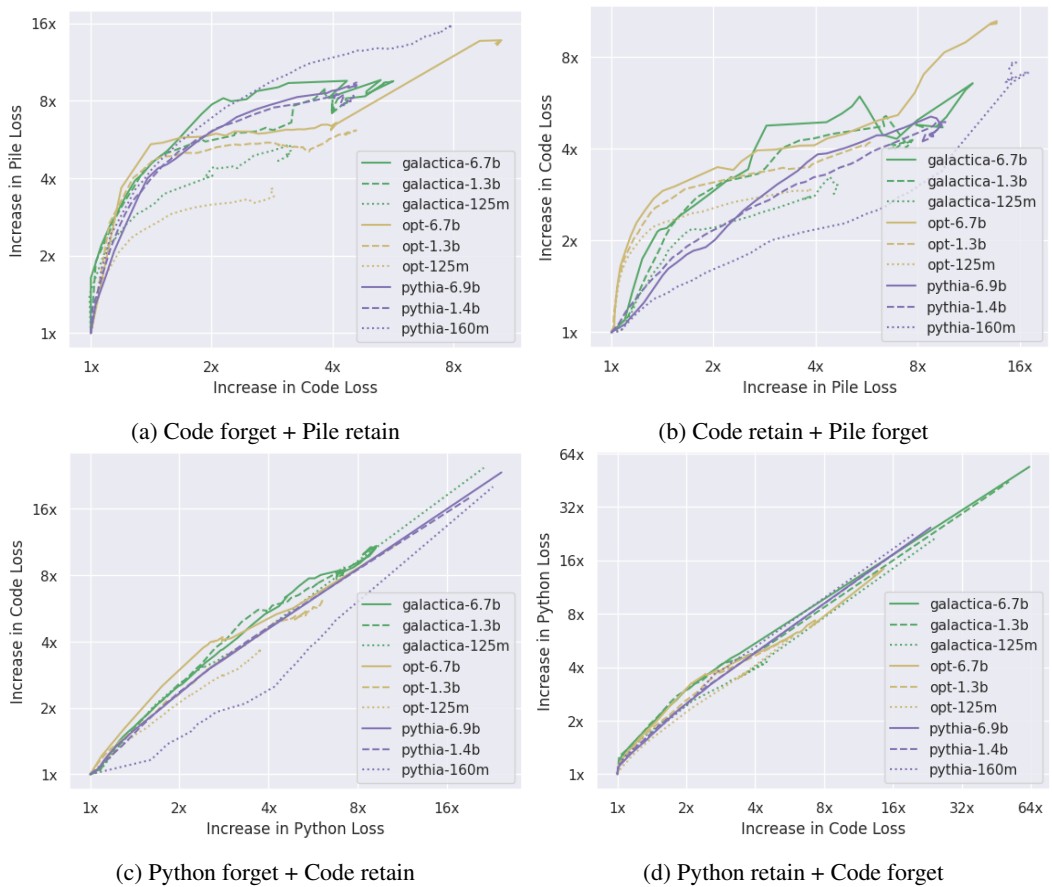

(a) Code forget + Pile retain

(b) Code retain + Pile forget

(c) Python forget + Code retain

(d) Python retain + Code forget

Figure 12: Loss Trajectories for OPT, Galactica and Pythia models.

### E.3 MODEL PRUNING SCORES FOR THE CASE OF THE REVERSE TASK (CODE RETAIN PILE FORGET)

Figure 13 is similar to Figure 4 with the main difference being that here we look at the reverse task (Code retain Pile forget).

We see in Figure 13 that the 'best random' performance is different between the two tasks (reversed vs unreversed task), and since we are getting the maximal difference, the metric is not symmetric for Code retain and Code forget. We also note that the ordering of metrics changes in this comparison (that is, FF abs is no longer the best performer). Finally, we note that some of the attention pruning methods work worse than random. The metric that seems to work most consistently well is mean absolute activation (abs).

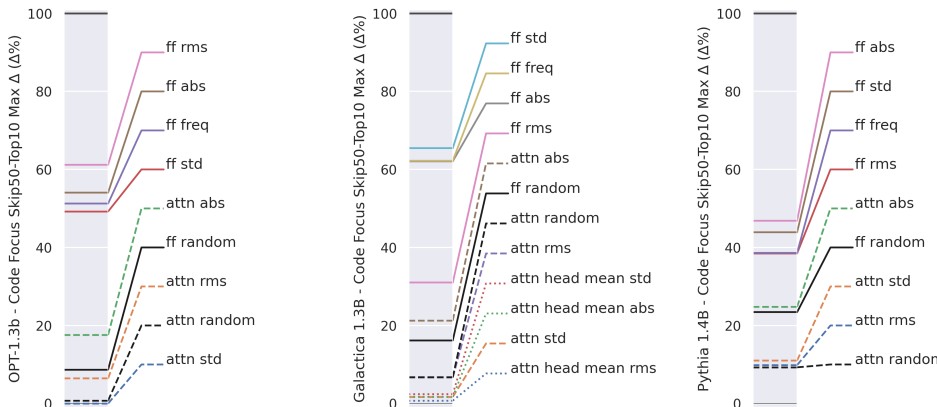

Figure 13: We evaluate methods for pruning OPT-1.3B (left), Galactica-1.3B (center) and Pythia-1.4B (right). We use various different importance functions (freq, abs, rms, std), on different regions of the model (feed-forward neurons, attention "value neurons" and attention heads). The graphs show the maximal difference between Skip50-Top10 accuracy in Pile and Skip50-Top10 accuracy in Code performance over pruning steps.

