# OpenReview forum: "Dissecting Language Models: Machine Unlearning via Selective Pruning"
_ICLR.cc/2024/Conference — Submitted to ICLR 2024_

### Official Review · Reviewer_Q3q4 · 2023-10-30

**Soundness:** 3 good
**Presentation:** 2 fair
**Contribution:** 3 good
**Rating:** 6
**Confidence:** 4

**Summary:**

This paper presents the application of pruning techniques to analyze neuron behavior within large language models. Introducing a method termed "selective pruning," the approach gauges the significance of each neuron based on its importance in both retained and forgotten datasets. From their experimental findings, the paper highlights that: (1) neurons exhibit high specialization, (2) larger models demonstrate more selective tendencies, and (3) feed-forward neurons show greater specialization compared to attention neurons.

**Strengths:**

1. This paper delves into an interesting topic not previously addressed: examining the behaviour of neurons in large language models to comprehend their functionality.
2. The article introduces a method termed "selective pruning" to undertake machine unlearning on large language models, subsequently employing it to analyze neuronal functions.
3. The research offers intriguing findings from its experiments, suggesting that neurons in FFN hold greater significance than in Attention module for specialized tasks. Such insights could potentially inspire further research and insights into large language models within the community.

**Weaknesses:**

1. The experimental results presented by the author do not fully support the conclusion this paper wants to draw. For example, the authors mentioned in the introduction that, "If capabilities can be separated on the level of neurons, then this can lead to modularity inside models." However, based on the experimental results, it appears that the entire LLM behaves highly in coupling and cannot be separated. For instance, in Figure 1(d), the performance loss on the 'code' dataset seems to be mirrored closely by a performance loss on the 'python' dataset, where the model drops close to the retain dataset and the forget dataset.

2. The paper lacks comprehensive and comparable comparisons with previous methods. The authors only show the results with one baseline method (Task Arithmetic), and the comparison does not seem equitable.  It's unclear from the presented data whether their approach outperforms the baseline method. Using varying scales for the reduction in perplexity complicates the evaluation,  and it's challenging to determine whether a reduction from 4.8 to 0.8 is significant, or if a drop from 2.2 to 0.3 is more significant.

3. Given that the experiments were solely conducted on datasets related to code, I am uncertain about the generalizability of the experimental results presented in the paper. For instance, the conclusion that FFN outperforms attention—might it be possible that a different task could yield an opposing conclusion.

4. The readability of the entire article is not good. For instance, in Section 3.1, the author describes the distribution characteristics of the "**attention pre-out neuron**" activations. However, the definition of this unfamiliar term, "attention pre-out neuron," is only introduced in Section 3.3. This leads to confusion for me when initially encountering the term. Additionally, the article frequently places experimental results in the appendices, referencing them in the main text and using the conclusion from these experiments in appendices to support further observation in the main text. This approach disrupts the reading flow, often requiring to flip back and forth for context. While thorough analyses and experiments are commendable, the structure of the article still needs further refinement to enhance its logical flow.

**Questions:**

1. Please answer the questions mentioned in Weaknesses.

2.  In section 3.2: As a baseline we also randomly pruned layers. Where is this baseline?

3. A prior study [1] demonstrated that, depending on the specific input, it's possible to achieve a high pruning ratio without negatively affecting performance and without the need for retraining. This suggests that there is redundancy in the neurons of the LLM when it's tasked with executing a singular function (equating a single sentence to a minor task, for instance). In light of these findings, what novel insights or observations does your paper offer in comparison to that study?

[1] Deja Vu: Contextual Sparsity for Efficient LLMs at Inference Time

---

> ### Author Response · Authors · 2023-11-16
>
> We would like to thank the reviewer for their helpful and thorough review, and for their positive comments. We’re glad you think some of our insights could potentially inspire further research and insights into large language models within the community. We address the concerns raised in the review below.
>
> > 1. However, based on the experimental results, it appears that the entire LLM behaves highly in coupling and cannot be separated. For instance, in Figure 1(d), the performance loss on the 'code' dataset seems to be mirrored closely by a performance loss on the 'python' dataset
>
> We think that the extent to which different capabilities are coupled or intertwined depends on the capabilities in question. By comparing the results in Figure 2(a) and 2(b) we see that ‘python’ ability and ‘code’ ability are more intertwined than ‘code’ ability and ‘pile’ (i.e. general text) ability. This is in line with what one might intuitively expect: coding ability in general and coding in the python programming language seem like similar skills.
>
> > 2. Using varying scales for the reduction in perplexity complicates the evaluation, and it's challenging to determine whether a reduction from 4.8 to 0.8 is significant, or if a drop from 2.2 to 0.3 is more significant.
>
> We agree that it is inconvenient that we were unable to replicate their results. In an attempt to resolve this we have contacted the authors in August, but unfortunately they have not replied.
>
> We have tried to further vary text generation parameters that were not stated in the paper, and by increasing the temperature and generation length, we managed to rerun the experiment to find pre-intervention toxicity levels that are closer to 4.8. The updated paper now reports a drop in toxic generations from 3.5% -> 0.3%.
>
> > 3. Given that the experiments were solely conducted on datasets related to code, I am uncertain about the generalizability of the experimental results presented in the paper. For instance, the conclusion that FFN outperforms attention—might it be possible that a different task could yield an opposing conclusion.
>
> We agree that the degree to which attention and feed forward pruning may be specialized to different tasks may differ. Thank you for pointing this out. We have updated the pdf to reflect this.
>
> > 4. The readability of the entire article is not good. For instance, in Section 3.1, the author describes the distribution characteristics of the "attention pre-out neuron" activations. However, the definition of this unfamiliar term, "attention pre-out neuron," is only introduced in Section 3.3. This leads to confusion for me when initially encountering the term. … While thorough analyses and experiments are commendable, the structure of the article still needs further refinement to enhance its logical flow.
>
> Thank you for the feedback. We have resolved the issue of using the term “attention pre-out neuron” before we explain the term. We have also made other updates to the paper to improve readability, such as improving the wording and adding a diagram describing our method on page 3.
>
> **Question 1 and Question 2:** Question 1 is addressed in the comments above and question 2 is answered in the common concerns section.
>
> **Question 3:**
>
> > A prior study [1] demonstrated that, depending on the specific input, it's possible to achieve a high pruning ratio without negatively affecting performance and without the need for retraining. … In light of these findings, what novel insights or observations does your paper offer in comparison to that study?
>
> This paper is very interesting indeed! We have now cited this paper. Their findings are similar to but more general than the ones in [A] which motivate our importance functions. The paper shows contextual sparsity per input sentence, whereas we show something similar per dataset.
>
> The paper shows that neurons specialize locally on the scale of tokens and sentences. We show that neurons specialize more globally on the scale of broader tasks and datasets.
>
> In Figure 1 (now Figure 2) we have now added imagenet data, and we think it is interesting that the three different pairs Pile-Coding, Coding-Python, Imagenet-Bird all have very different amounts of separability. This shows that the separability of neurons is task dependent.
>
> [A] MoEfication: Transformer Feed-forward Layers are Mixtures of Experts, Zhang et al. https://arxiv.org/abs/2110.01786
>
> ### **Conclusion**
>
> We hope the above clarifications and additional results have changed your view on the paper and successfully addressed all the questions and limitations you mentioned. Please let us know if there are other concerns stopping you from increasing your score and recommending acceptance of the paper.

---

> > ### Comment · Reviewer_Q3q4 · 2023-11-21
> >
> > Thanks for your detailed response. I would increase my score to 6.

---

### Official Review · Reviewer_sDUD · 2023-10-30

**Soundness:** 2 fair
**Presentation:** 3 good
**Contribution:** 2 fair
**Rating:** 5
**Confidence:** 4

**Summary:**

This paper proposes an unlearning algorithm based on pruning for removing ‘capabilities’ from pretrained language models. Specifically, they provide different definitions for quantifying the importance of a neuron for a particular dataset based on the value of its activation on that the points in that dataset. Then, they define a ‘score’ for a neuron as the ratio of importance of that neuron for the retain versus the forget datasets. They prune a chosen percentage of nodes from a ranked list according to this score. Empirically, they experiment in a setting where a general purpose model is caused to forget its ‘coding ability’ (and the reverse of forgetting all but its coding ability). They also look at a more finegrained scenario of removing python coding ability while retaining the ability to code in other languages (and its reverse). The way that forgetting is measured is by inspecting the relative drop in accuracy on the forget set (relative to the relative drop on the retain set). My understanding is that it is desired according to this evaluation to have a large drop in the forget accuracy without having a large drop in the retain accuracy. They investigate empirically these trade-off curves obtained by their method in different types of language models. Then, on a different task/dataset (where the goal is to remove toxicity from a trained language model), they compare against one baseline and claim that they get similar results.

**Strengths:**

- the paper studies an interesting and important problem
- the proposed method is a reasonable idea and well motivated
- the proposed method is efficient and requires no gradient updates
- the paper is for the most part well-written (though see some exceptions below)

**Weaknesses:**

- the related work section is weak, missing a lot of literature both from unlearning and pruning. For example, [A-E] are recent papers on unlearning, with [A, B] particularly related to the methodology of this paper (see References below). I’m less familiar with the sparsity and pruning literature but the authors should conduct a thorough review.
- the authors compare against only one baseline, and only for one task. Other common baselines include finetuning on the retain set, gradient ascent on the forget set, and comparing against other recent unlearning methods is also important (see the references below)
- the particular problem setting of unlearning is not clearly defined. What defines successful unlearning here? In section 3 the authors describe the forget set as the “task or dataset that we aim to reduce performance on”. This isn’t precise enough. How much do we want to reduce performance? Is it that the greater the reduction, the better? For context, unlearning papers (e.g. Golatkar et al, which the authors cite) usually consider that the accuracy on the forget set should be reduced only up to a reference point and no further (where the reference point is given by the oracle unlearning algorithm of retraining the model from scratch without the forget set). Is the setup here different, and if so, how is the goal defined in this case?
- [clarity] It seems that ‘task’ and ‘dataset’ are used interchangeably in the paper which causes confusion. For example, ‘the task that we are optimizing for as the retain dataset’. To me, a ‘task’ includes a particular training objective whereas ‘dataset’ refers to raw data.
- [clarity] fundamental problem setting details are missing. For example, were Pile and Code included in the dataset that the various language models were trained on? If not, the terms “forgetting” or “unlearning” may be ill-suited for this application (as they usually refer to forget parts of the training dataset). At the very least, the problem setting targeted by this paper should be clearly defined.
- [soundness]: can decreased performance (accuracy, perplexity) on a particular dataset support claims of removal of a capability? Generally, ‘capability removal’ is not precisely defined.
- [soundness]: when it comes to forgetting or unlearning, several metrics have been proposed by the community to measure this. Simply inspecting the accuracy / perplexity is likely a poor proxy for forgetting quality. For example, Membership Inference Attacks are an important category of methods (see e.g. Golatkar et al and reference [E] below). Are these not applicable here. If not then why not?
- [presentation, soundness] the authors use the term “selective” to describe an unlearning method, without defining this term clearly in this context. My understanding of what “selective” means in this context is the ability of reducing accuracy / performance on the forget dataset without (really) damaging the accuracy / performance on the retain dataset. If my understanding is correct, I don’t agree with the claim that Figure 1a shows that the larger the model, the more selective it is. I can see this being true for the Opt family but not the Pythia family, for example.
- [presentation] In the paragraph under Definition 1, the authors give intuition for the different influence functions but they omit I_{abs}. Please add.
- [presentation] Above Table 1, the authors describe the models they use (which correspond to columns in Table 1) but they omit Roberta. Please add.
- [presentation] “As a baseline, we also randomly pruned layers” – the authors claim this but I don’t see this baseline in their experiments (at least in the main paper), unless I’m missing it.
- [empirical results] It would greatly strengthen the paper to conduct analyses of: the effect of the number of pruning iterations, the effect of the % pruned (in each iteration or overall), the effect of the choice of the influence function. It sounds like the authors have some results on some of these in the Appendix but not all of them. It would also be great to summarize in the main paper all of the findings (so that one doesn’t need to read the entire Appendix).
- [empirical results] please include confidence intervals in all tables and e.g. in Figure 3. It is currently challenging to tell if the differences are significant
- [empirical results] why is there such a large difference between “Base (quoted)” and “Base (replicated)” in Table 3? This makes me concerned about whether “Task Arithmetic (quoted)” and “Pruned” are comparable.

References
=========
- [A] Fast Machine Unlearning Without Retraining Through Selective Synaptic Dampening. Foster et al 2023.
- [B] Model Sparsity Can Simplify Machine Unlearning. Jia et al. NeurIPS 2023.
- [C] Unrolling SGD: Understanding Factors Influencing Machine Unlearning. Thudi et al.
- [D] Prompt Certified Machine Unlearning with Randomized Gradient Smoothing and Quantization. Zhang et al. NeurIPS 2022.
- [E] Towards Unbounded Machine Unlearning. Kurmanji et al. NeurIPS 2023.

**Questions:**

- the authors claim that their method is specifically designed for LLMs but it’s unclear to me why that’s the case. Can’t this method be applied out of the box e.g. to vision transformers? If not, then why not?
- In Section 3.1, the authors discuss some observations about the distributions of activations that motivated their use of importance functions. Which dataset / setting were these observations made in? Have the authors made an effort to confirm that they are generalizable beyond a certain setting?
- I don’t really understand the statement that “We also do not want to directly modify specific dimensions of the embedding space”. Did not understand the provided rationale. If it’s not the right level of granularity for pruning, as the authors claim, would the proposed scoring function not capture this? If not, does this suggest we need to design better scoring functions?
- In the Discussion, the authors hypothesize that their method is more likely to “actually remove the undesired behaviour” compared to other unlearning methods. Similarly, in the Broader Impacts section, they claim that (compared to other unlearning methods) their method is unlikely to generate systems that are more harmful than the base model. What is the evidence used for making these claims?

---

> ### Author Response · Authors · 2023-11-16
>
> We thank the reviewer for their detailed and helpful review, and for their positive comments. We’re glad you find we study an interesting and important problem and that the proposed method is well motivated, efficient and requires no gradient updates.
>
> > [A-E] are recent papers on unlearning
>
> > Can’t this method be applied out of the box e.g. to vision transformers?
>
> [A] This Arxiv paper from August 15th is related and interesting!
> In a quick trial we applied our method to the Vision Transformer (ViT) mushroom (MR) row in their Table 2. On the test split (100 datapoints) of CIFAR100 we find a forget drop from 72±8% to 0±3% and a retain drop from 90.0±0.3% to 89.5±0.3%.
> [A-E] Cited.
>
> > Common baselines include finetuning on the retain set, gradient ascent on the forget set
>
> Reference [B] states that fine-tuning and gradient ascent based machine unlearning methods cost between 2% and 6% the cost of retraining a model. For reference, training Llama-2-70b used 1.7million GPU hours.
>
> > How much do we want to reduce performance?
>
> While we agree that retraining is a good standard, retraining an LLM is too expensive. We simply try to maximise drop in forget while keeping performance in retain, seen with a "curvier" graph being better. We do not insist on a specific goal, but make a cheap reference, and allow the user to decide where they would like to be on the trade-off between a high retain & low forget accuracy.
>
> > Were Pile and Code included in the dataset that the various language models were trained on? If not, the terms “forgetting” or “unlearning” may be ill-suited for this application.
>
> It's uncertain which specific data points were used in the training of all models (e.g., LLaMA 2). Some models, such as Pythia and OPT, do incorporate parts of the Pile dataset.
>
> The terms 'retain' and 'forget' are existing terminologies that mostly capture the meaning we want. Though it may slightly deviate from normal usage, inventing new terminology would likely be more confusing. Rather than referring to the forgetting of specific instances encountered during training, we're focusing on the loss or retention of underlying skills as exemplified by a dataset.
> Our goal is to remove harmful behavior as exemplified by a (forget) dataset. For example, GPT might not have seen “To make a bomb you need x, which you can get at location y”, but the model may still output such strings.
>
> > Membership Inference Attacks are an important category of methods...
>
> In this paper we have tried to show that harmful behavior can be reduced by applying selective pruning. We are less focused on removing specific datapoints from the model than we are in removing the model’s ability to generate code or toxic data. Studying membership inference attacks (which requires full knowledge of training data) is beyond the scope of our current analysis.
>
> > I don’t agree … the larger the model, the more selective it is.
>
> Thank you for pointing this out. We agree and we have updated the pdf to reflect this.
>
> > [presentation] … I_{abs} … Table 1,
>
> Thanks! Added.
>
> > It would greatly strengthen the paper to conduct analyses of: the effect of the number of pruning iterations, the effect of the % pruned
> (in each iteration or overall), the effect of the choice of the influence function.
>
> We have chosen not to do a grid-search for the hyperparameter “percentage pruned per iteration”, but instead report on all the hyper-parameters that we tried.
> The effect of how much has been pruned overall can be seen in Figure 2. Every dot here reflects another pruning step. The more pruning steps the bigger the reduction in accuracy.
> Figure 4 shows the effect of the choice of influence function.
>
> > please include confidence intervals
>
> In our tables, we apply the 'significant digits' convention, where we round our values to the most relevant figure and omit digits that represent a substantial error margin. We have taken the suggestion to add error bars to Figure 4.
>
> > Which dataset / setting were these observations made in?
>
> These were taken using OPT on the Pile dataset, we have verified the observations for Pythia and LLaMA.
>
> > “We also do not want to directly modify specific dimensions of the embedding space”
>
> After every layer, the model writes into the residual stream which has the same dimension as the embedding space (and which at the last layer turns into the model output after the unembedding). Investigating how to alter these dimensions that are being continuously read from and written into would be an interesting investigation, but is beyond the scope of this paper.
>
> > In the Discussion, the authors hypothesize that their method is more likely to “actually remove the undesired behaviour” compared to other unlearning methods...
>
> We compare to other model control methods more broadly, many of which have been shown to be brittle under jailbreaking and finetuning. We hypothesize that (our) machine unlearning methods are less brittle than these control methods.

---

> > ### Comment · Reviewer_sDUD · 2023-11-21
> > **thank you for the responses**
> >
> > Hi authors,
> >
> > Thank you for your responses! here are some additional thoughts after reading the rebuttal and other reviews:
> >
> > re: "We simply try to maximise drop in forget while keeping performance in retain, seen with a "curvier" graph being better. We do not insist on a specific goal, ..." -- it's not clear to me what sort of application this setup targets exactly. In the intro, the authors motivate unlearning by the need to protect against misuse or misalignment of LLMs or remove sensitive user information. Without a clear reference point for e.g. how much the accuracy should drop on a "task", it seems impossible to assess if these desiderata can be reached. For example, when it comes to protecting user privacy, there is discussion in the literature that reducing the accuracy on such user data 'too much' might 'backfire' and make those data points more vulnerable to e.g. membership inference attacks (which the authors argue in their rebuttal is beyond the scope of their exploration). I think it would be useful for the authors to specifically motivate their problem setup by some realistic scenario and adopt metrics that reflect that.
> >
> > re: other baselines: i don't think it's valid to dismiss comparing against any other baselines or methods from the unlearning literature due to claims that they are too computationally expensive. These comparisons could be run, for instance, on smaller datasets (e.g. the vision dataset included in the rebuttal -- which btw is a great addition!). Generally, comparing to prior work is a crucial step to understand how the proposed method differs and inform practitioners when they should use it over other methods.
> >
> > re: "We compare to other model control methods more broadly, many of which have been shown to be brittle under jailbreaking and finetuning" -- could you please include some reference for this? in addition, i'm curious about this hypothesis that the proposed method is less brittle. what is the reasoning and evidence there?
> >
> > overall, I find this work really interesting and the proposed method seems promising. but I feel that another round of reviews is needed to clarify the motivation / application scenario, and expand the experimental section to include comparisons to previous works.

---

> > > ### Author Response · Authors · 2023-11-22
> > >
> > > > Without a clear reference point for e.g. how much the accuracy should drop on a "task", it seems impossible to assess if these desiderata can be reached. … I think it would be useful for the authors to specifically motivate their problem setup by some realistic scenario and adopt metrics that reflect that.
> > >
> > > We agree this is an important issue, and we mostly envision that a good reference point would be evaluating generative tasks such as writing a working program or correct instructions. We would want our method to impair ability to write a working program or provide accurate instructions for building a bomb. Most of these assessments, however, are only viable on the largest, state-of-the-art models (e.g. we were only able to get decent results on MMLU for LLAMA 2), making it more difficult to run experiments. We aimed to give proxies for this in our experiments.
> > >
> > > Also note that there are ethical constraints in what experiments and benchmarks can be set up without the potential for societal harm. For example, providing metrics for “how easy is it to get a model to provide instructions for making a bomb” is likely to result in the publishing of procedures that are highly effective in eliciting this dangerous information.
> > >
> > > We believe reducing toxic generation in language models (while retaining ability on MMLU) showcases the method is applicable in real-world contexts, but we agree more research could be done.
> > >
> > > > These comparisons could be run, for instance, on smaller datasets (e.g. the vision dataset included in the rebuttal -- which btw is a great addition!). Generally, comparing to prior work is a crucial step to understand how the proposed method differs and inform practitioners when they should use it over other methods.
> > >
> > > We acknowledge the suggestion, and have updated the paper to add comparisons to previous work for vision transformers on CIFAR100. Specifically, we compare to: retraining; finetuning; the ‘incompetent teacher’ method; UNSIR; amnesiac and selective synaptic dampening. We have also included a membership inference attack (MIA) metric.
> > >
> > > See Table 4 on page 8 of the newly updated pdf.
> > >
> > > > "We compare to other model control methods more broadly, many of which have been shown to be brittle under jailbreaking and finetuning" – could you please include some reference for this?
> > >
> > > RLHF [1] (a control method) guard rails have shown to be fragile to jailbreaking [2] and finetuning [3]. In the running example, this is possible because the information about how to build a bomb is still in the model. Activation addition [4] adds a vector to activations during inference time, this method can be undone by again subtracting the vector from the activations.
> > > Our method (like other machine unlearning methods) is an attempt to remove much of that (bomb building) information so that even if one unlearns the guard rails, one can still not learn how to make the bomb.
> > >
> > > [1] Deep reinforcement learning from human preferences, Christiano et al. https://arxiv.org/abs/1706.03741
> > >
> > > [2] Shen, Xinyue, et al. "" Do Anything Now": Characterizing and Evaluating In-The-Wild Jailbreak Prompts on Large Language Models." arXiv preprint arXiv:2308.03825 (2023).
> > >
> > > [3] LoRA Fine-tuning Efficiently Undoes Safety Training in Llama 2-Chat 70B, Lermen et al. https://arxiv.org/abs/2310.20624
> > >
> > > [4] Turner, Alex, et al. "Activation addition: Steering language models without optimization." arXiv preprint arXiv:2308.10248 (2023).

---

### Official Review · Reviewer_9FMg · 2023-11-01

**Soundness:** 3 good
**Presentation:** 3 good
**Contribution:** 3 good
**Rating:** 6
**Confidence:** 3

**Summary:**

This work presents a novel selective pruning method in order to allow trained models to 'unlearn' specific capabilities. Related work and the details of the method are explicated. Experiments on different models with different sizes are presented on two data splits (code/pile and code/python) showing generally good performance at 'unlearning' the forgetting dataset and retaining the other. Analysis experiments compare the efficacy of pruning feed-forwards vs attention neurons. Experiments on toxicity show good results for the method on allowing the unlearning of toxic text.

**Strengths:**

The paper is clearly written. The experiments are thorough, and the presented results convincingly demonstrate the utility of the method. The results on toxicity are particularly nice, as this is a highly relevant domain for which related techniques are well-motivated. The presented method is novel.

**Weaknesses:**

In Limitations: "Our method can only be applied to remove a capability when that capability is neatly captured by a
dataset." The authors rightly point out that this method of evaluation of unle'arning is dependent upon dataset-level perplexity. The extent to which this metric for any constructed dataset sufficiently "neatly captures" whatever capability is desired to be forgotten is difficult to asses without further analysis not presented in this work. It may be the case that for the practical scenarios which motivate the method in the first place, no such "neat" dataset is possible to produce. It is fine for this analysis to be out off scope of this work.

Given this evaluative dependence upon specific datasets, it would significantly strengthen the results of the paper to present more diverse experiments. While it is nice to have many different models at many different sizes, there does not seem to be much addition knowledge gleaned from that diversity, whereas having more tasks would demonstrate a broader efficacy across a more speculative dimension.

In this paper, the authors present experiments pruning either the feed-forward or the attention blocks, and leave "Embedding, Positional
Embedding and Output Unembedding unmodified." (3.3). This is a significant constriction of the application of the method which is not more than intuitively justified. Further experiments, even just to show that this restriction is well-motivated, would strengthen the work.

**Questions:**

See weaknesses.

In Table 3: it would help readability to label "Task Arithmetic (quoted)" as being a finetuning task. Also, the gap between the baseline (quoted) and (replicated) is fairly large, which makes actually comparing the finetuning vs presented pruning method difficuly. Would it be possible to replicate the task-arithmetic finetuning? That would significantly improve the comparability of these results.


small issue:
The details of iterative pruning are not specified. It seems like it would be important to the function of the method to set the proportion of nodes pruned per iteration well, but this is not discussed. Even if this is not and important hyper-parameter of the method, a clearer explanation of the iterative pruning method is necessary to fully elucidate the applied method.

6.3: If an LLM were trained not to answer questions about a dangerous topic, say, bomb-building, could the presented method not be used to unlearn that guardrail? I don't think this is a concern specific to the presented method, but I do not follow why this method is less likely to generate harmful systems than other methods. Can the authors clarify?

Nit:
in discussion: hypothesise -> hypothesize

---

> ### Author Response · Authors · 2023-11-16
>
> We would like to thank the reviewer for their detailed review, and for their positive comments. We’re glad you find the presented method novel and our experiments  thorough, and that the presented results convincingly demonstrate the utility of the method. We are also happy you found toxicity removal a highly relevant domain. We address the concerns raised in the review below.
>
> > Given this evaluative dependence upon specific datasets, it would significantly strengthen the results of the paper to present more diverse experiments. While it is nice to have many different models at many different sizes, there does not seem to be much additional knowledge gleaned from that diversity, whereas having more tasks would demonstrate a broader efficacy across a more speculative dimension.
>
> We have now included results for an image classification task. Please see them in the updated version of Figure 1 (now moved to be Figure 2) of the updated pdf.
>
> > In this paper, the authors present experiments pruning either the feed-forward or the attention blocks, and leave "Embedding, Positional Embedding and Output Unembedding unmodified." (3.3). This is a significant constriction of the application of the method which is not more than intuitively justified.
>
> The Embedding, Positional Embedding and Output Unembedding are one layer / weight matrix each. Excluding three layers from pruning out of the 30-100 layers that LLMs typically have seems like a minor constriction. The embedding and output unembedding exist to translate from the input (or output) tokens to an internal model representation and almost function like a dictionary. In LLama the positional embedding is not even learned, but is instead hard-coded. This positional embeddings seem like they would be essential for most tasks that one might think of.
>
> > In Table 3: it would help readability to label "Task Arithmetic (quoted)" as being a finetuning task.
>
> Thank you for the suggestion! We have updated the label.
>
> > The details of iterative pruning are not specified. It seems like it would be important to the function of the method to set the proportion of nodes pruned per iteration well, but this is not discussed.
>
> We have added some clarification in Section 3.2 in the updated pdf. We have not done any hyper-parameter tuning for the proportion of nodes pruned per iteration.
>
> > 6.3: If an LLM were trained not to answer questions about a dangerous topic, say, bomb-building, could the presented method not be used to unlearn that guardrail? I don't think this is a concern specific to the presented method, but I do not follow why this method is less likely to generate harmful systems than other methods. Can the authors clarify?
>
> Something that was not clear in our original formulation is that we think machine unlearning methods (such as ours) will be more thorough than other model control methods.
>
> For example, RLHF (a control method) guard rails are fragile to things like jailbreaking [1] and finetuning [2]. This is possible because the information about how to build a bomb is still in the model. This method is an attempt to remove much of that information so that even if one unlearns the guard rails, one can still not learn how to make the bomb.
>
> [1] Deep reinforcement learning from human preferences, Christiano et al. https://arxiv.org/abs/1706.03741
>
> [2] LoRA Fine-tuning Efficiently Undoes Safety Training in Llama 2-Chat 70B, Lermen et al. https://arxiv.org/abs/2310.20624
>
> ### **Conclusion**
>
> We hope the above clarifications and additional results have changed your view on the paper and successfully addressed all the questions and limitations you mentioned. Please let us know if there are other concerns stopping you from increasing your score.

---

### Official Review · Reviewer_q42J · 2023-11-01

**Soundness:** 3 good
**Presentation:** 3 good
**Contribution:** 3 good
**Rating:** 6
**Confidence:** 4

**Summary:**

This article presented an approach for the targeted removal of neurons, which is based on their comparative significance across two datasets. This technique of machine unlearning demonstrates its effectiveness through the quantifiable decrease in accuracy differentials and perplexity measurements. Additionally, it establishes a cost-effective foundation for forthcoming research comparisons. Its theory posits that the approach is more inclined to eliminate undesired model behaviors, as opposed to merely concealing them, in contrast to fine-tuning.
This approach is a compute- and data-efficient method for identifying and removing neurons that enable specific behaviours. The findings of this method reveals that both feed-forward and attention neurons in LLMs are specialized; that is, for specific tasks, certain neurons are more crucial than others.

**Strengths:**

1- The model presented here tackles an intriguing and complex issue within large language models (LLMs), focusing on the significance scores assigned to individual neurons with respect to a specific target dataset.

2- The concept of machine unlearning is implemented across both types of neural networks, including feedforward and attention-layer-based networks. By eliminating unwanted neurons for any given target dataset, the process is swift and leads to a reduction in the network's computational load.

3- The efficacy of the unlearning process is demonstrated through experiments conducted on three distinct datasets.

**Weaknesses:**

1- The proposed model can effectively eliminate the information captured by the target dataset. However, it is unable to unlearn knowledge that lies beyond the representation of the datasets. Please provide a clear justification.

2- Including a visual representation of the suggested concept could enhance comprehension. Thus, kindly incorporate a diagram that presents a general outline of the proposed concept.

3- The evaluation of the proposed model focuses on text datasets, yet the experiments for the image dataset are absent. Implementing machine unlearning for the image dataset would be highly beneficial.

**Questions:**

Please address all the question raised in the weaknesses section.

---

> ### Author Response · Authors · 2023-11-16
>
> We would like to thank the reviewer for their helpful review, and for their positive comments. We’re glad you found the goal of removing capabilities from LLMs an intriguing and complex issue, and our method a cost-effective foundation for forthcoming research comparisons, the efficacy of which was demonstrated through experiments on three distinct datasets. We address the concerns raised in the review below.
>
> > 1- The proposed model can effectively eliminate the information captured by the target dataset. However, it is unable to unlearn knowledge that lies beyond the representation of the datasets. Please provide a clear justification.
>
> Yes we think this is an important question! However, we think that the general question of capturing (un)desired behaviour in a dataset is one of the key problems in machine learning. It is outside the scope of this paper to solve this problem here.
>
> > 2- Including a visual representation of the suggested concept could enhance comprehension. Thus, kindly incorporate a diagram that presents a general outline of the proposed concept.
>
> We have now added a visual representation (new Figure 1) of selective pruning to Section 3.2 on page 3 of our updated pdf.
>
> > 3- The evaluation of the proposed model focuses on text datasets, yet the experiments for the image dataset are absent. Implementing machine unlearning for the image dataset would be highly beneficial.
>
> We have taken your suggestion to implement selective pruning for an image dataset on board! Please see our updated Figure 1 (now Figure 2) in the updated pdf.
>
> ### **Conclusion**
>
> We hope the above clarifications and additional results have changed your view on the paper and successfully addressed all the questions and limitations you mentioned. Please let us know if there are other concerns stopping you from increasing your score.

---

### Author Response · Authors · 2023-11-16

We thank all reviewers for their time and effort in writing their reviews. In general, reviewers found the goal of removing capabilities from LLMs interesting and important, and our method well-motivated and efficient.


Below we address some concerns shared by multiple reviewers.

*Reviewer 1, 2 and 4 ask for experiments with non-coding datasets.*

We have now applied our method to vision transformer models, where we use the Bird class as a forget dataset, while retaining the ability to classify all other classes. We have added our results to our updated Figure 1 (now Figure 2) on page 6 of the updated pdf.

*Reviewer 3 and 4 ask why is there such a large difference between “Base (quoted)” and “Base (replicated)” in Table 3?*

We agree that it is inconvenient that we were unable to replicate their results to full precision. In an attempt to resolve this, we contacted the authors in August, but unfortunately they have not replied.

We have tried to further vary text generation parameters that were not stated in the paper, and by increasing the temperature and generation length, we managed to rerun the experiment to find pre-intervention toxicity levels that are closer to 4.8. The updated paper now reports a drop in toxic generations from 3.5% to 0.3%.

*Reviewer 3 and 4 ask where we show the baseline of also randomly pruned layers.*

We show the randomly pruned layers in Figure 3. We have chosen not to add them to Figure 1 (now Figure 2), because the randomly pruned baseline, though similar, is different for each model, and this figure contains data for ten different models.

---

### Meta-Review · Area_Chair_Die6 · 2023-12-07

**Metareview:**

This paper proposes selective pruning/removal of neurons based on various importance functions w.r.t. the retained and forget datasets.


**STRENGTHS**

(1) The problem is interesting and important.

(2) The proposed method is simple, efficient, and does not need gradient updates.

(3) The experimental findings are interesting.


**WEAKNESSES**

A number of major concerns remain and recommendations are provided to improve the work in this paper, as discussed below.

(1) As this is a purely empirical paper, several reviewers have raised the need of a more comprehensive experimental evaluation and analysis over a wider range of datasets in different domains (beyond what are presented in the current draft and rebuttal) and in comparison to existing unlearning works both quantitatively and qualitatively (pros and cons); the newly added Table 4 does not have an accompanying analysis of the results. A sensitivity analysis of the various hyperparameters (in particular, proportion of nodes pruned per iteration) needs to be provided too.

(2) Despite the discussion in the rebuttal, it remains unclear whether the unlearning metric/goal proposed by the authors (i.e., maximise drop in forget while keeping performance in retain) is reasonable. For example, if there are multiple similar copies of a code contributed by different people and only one needs to be removed, does it then fit into the authors' motivating scenario? In this regard, the authors need to provide a clearer motivation with use-cases such as some of those provided in the response, as well as those "non"-use-cases where this work should not be considered, hence forming the limitations.

(3) The related work needs to be beefed up considerably, especially in the literature on unlearning and pruning, as mentioned by a reviewer. For example, isn't it possible to use the proposed importance functions together with various existing pruning techniques?


Due to the above major concerns, this work requires a major revision. The authors are strongly encouraged to revise their paper carefully to address the above as well as the additional reviewers' concerns well.

**Justification For Why Not Higher Score:**

A number of major concerns remain, as discussed in the meta-review. There is also no strong advocate for the acceptance of this work.

**Justification For Why Not Lower Score:**

N/A.

---

### Decision · Program_Chairs · 2024-01-16

Reject